# HT-Step: Aligning Instructional Articles with How-To Videos

**Triantafyllos Afouras**[*]   **Effrosyni Mavroudi**[*]
**Tushar Nagarajan**   **Huiyu Wang**   **Lorenzo Torresani**
FAIR, Meta
{afourast, emavroudi, tusharn, huiyuw, torresani}@meta.com

## Abstract

We introduce HT-Step, a large-scale dataset containing temporal annotations of instructional article steps in cooking videos. It includes 116k segment-level annotations over 20k narrated videos (approximately 2.1k hours) of the HowTo100M dataset. Each annotation provides a temporal interval, and a categorical step label from a taxonomy of $4,958$ unique steps automatically mined from wikiHow articles which include rich descriptions of each step. Our dataset significantly surpasses existing labeled step datasets in terms of scale, number of tasks, and richness of natural language step descriptions. Based on these annotations, we introduce a strongly supervised benchmark for aligning instructional articles with how-to videos and present a comprehensive evaluation of baseline methods for this task. By publicly releasing these annotations and defining rigorous evaluation protocols and metrics, we hope to significantly accelerate research in the field of procedural activity understanding.

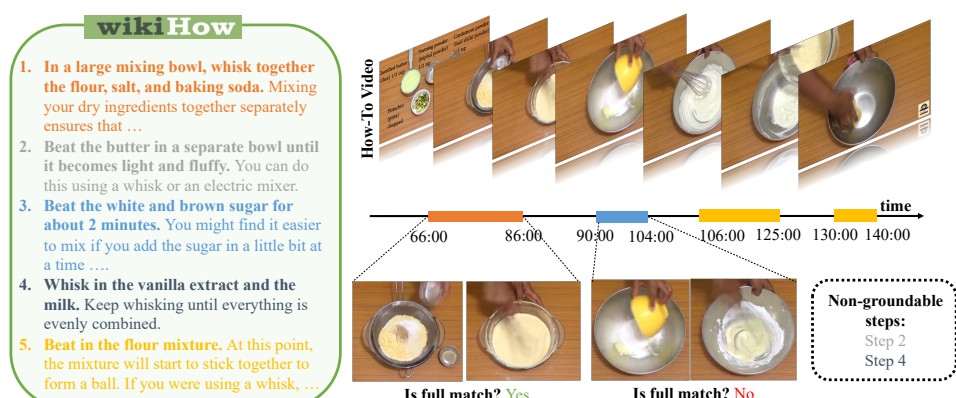

Figure 1: HT-Step is a dataset of annotations aimed towards instructional video-article alignment. Given an untrimmed how-to video and an instructional article containing a list of step descriptions, the goal is to temporally localize the steps shown in the video and reject the ones not demonstrated. Our dataset includes over 116k segment-level annotations for this task, as well as labels indicating whether a temporal segment is a *full* or a *partial* match to the step, i.e., whether the segment demonstrates all substeps and ingredients mentioned in the step headline (shown in **bold**).

---

[*]equal contribution

37th Conference on Neural Information Processing Systems (NeurIPS 2023) Track on Datasets and Benchmarks.

# 1 Introduction

Instructional videos have become a popular way to learn or improve our skills, providing an entertaining and dynamic alternative to traditional written manuals. For example, how-to videos are listed as one of the top-four watched categories on the YouTube platform [1], where viewers can access narrated visual demonstrations for a wide range of activities, from cooking to crafts and DIY projects. In addition to being a valuable learning medium for humans, how-to videos are a great source of training data for computer vision models [6, 25, 30]. Since many of these instructional videos focus on complex activities entailing a sequence of steps, these repositories offer great potential for learning task graphs [47] and training procedural activity models [42, 50]. However, although the videos are often accompanied by informative closed captions or ASR transcriptions, they lack structured step annotations that clearly denote which video segments correspond to what steps of the activity.

An alternative source of information for acquiring new skills is instructional article collections, such as wikiHow [20]. These typically provide a list of procedural steps and have the advantage of being more comprehensive, incorporating variations and detailed explanations. Although they often contain still-pictures or graphic illustrations, they lack the dynamic demonstrations offered by videos.

In this work we introduce HT-Step, the largest existing dataset with labeled step segments on instructional videos. We leverage wikiHow articles [20] to guide the annotation on a large subset of the open-sourced HowTo100M dataset [31]. These cross-dataset associations provide us with a taxonomy of steps well fitting the data and with an automatic mechanism to identify the set of potential steps represented in each individual video. Through a meticulous process of manual labeling by professional annotators we obtain 116k temporal step labels over 20k videos (approximately 2.1k hours). Our benchmark significantly surpasses the largest existing datasets of labeled steps [41, 50] in procedural videos along multiple axes: scale ($2.6\times$ the number of segments annotated, $4.8\times$ the number of video hours), taxonomy granularity ($7\times$ the number of steps), and scope ($2\times$ the number of activities).

We share these annotations with the research community and present a benchmark for the task of temporal article grounding. Given as input a how-to video and an instructional article listing a sequence of potential steps, the objective is to temporally ground the subset of steps that are demonstrated in the video and to detect outliers (i.e., the steps not executed). We note that this task differs from traditional temporal grounding task which require localizing a *single* query that is known to be *present* in the video. Instead, in our problem setting the system is given the *complete sequence* of potential steps. We believe that this formulation may encourage the design of methods that can perform global reasoning over all the steps and take full advantage of ordering constraints and sequence priors to disambiguate the grounding. Furthermore, on average about $58\%$ of the steps listed in the associated wikiHow article are *not* demonstrated in the given video. This implies that the model must have strong capability to recognize the steps that are not groundable. Finally, we include in our benchmark a test set over *unseen* tasks, i.e., activities that are not represented in the training set. Traditional activity detection methods requiring a taxonomy known a priori cannot generalize to unseen activities. By introducing this setting we hope to promote the development of language-based temporal alignment models that can learn to ground the rich textual descriptions of steps in instructional articles even for never-seen tasks.

In summary, our contributions are the following: (1) We introduce a large collection of step annotations on videos from the popular HowTo100M dataset. (2) We propose a scalable pipeline for annotating procedural videos that leverages an automatic strategy for associating potential steps to video and for inferring the step taxonomy. (3) We introduce splits, evaluation protocols and metrics that evaluate the ability of models to predict the temporal extent of steps in videos, and reject steps that are not visually groundable. (4) We train and evaluate state-of-the-art approaches and baselines that tackle our task from three different angles: activity detection, single-sentence temporal grounding, and temporal article grounding. (5) We discuss properties and unique aspects of HT-Step through comprehensive analyses and experiments.

# 2 Related Work

**Procedural and Instructional Video Datasets.** Procedural videos portray humans performing sequential steps in some constrained yet non-unique order, with the aim of achieving a specific goal, such as preparing a dish. Understanding the content of such videos has been an active area of

Table 1: Comparison between HT-Step and existing procedural video understanding (*top*), temporal grounding (*middle*), and instructional video (*bottom*) datasets. Note that HT-Step seen validation and test sets introduced in [28] are subsumed by HT-Step.

| Dataset | Duration (h) | # Videos | # Segments | # Activities | Domain |
|---|---|---|---|---|---|
| *Procedural Activity Datasets* | | | | | |
| MPII [35] | 10 | 44 | 5.6k | - | cooking |
| 50Salads [40] | 4.5 | 50 | 899 | - | cooking |
| TACoS [34] | ∼ 600 | 127 | 18.8k | - | cooking |
| Breakfast [22] | 77 | 2k | 8.5k | 10 | cooking |
| Ikea-FA [16] | 4 | 101 | 1.9k | 1 | furniture |
| EPIC-KITCHENS [9] | 100 | 700 | 90k | - | cooking |
| EgoProcel [3] | 62 | 298 | 1k | 16 | multiple |
| Assembly101 [36] | 513 | 4.3k | 1M | 101 | assembly |
| *Temporal Grounding Datasets* | | | | | |
| ActivityNet Captions [21] | 849 | 20k | 100k | - | multiple |
| Charades-STA [14] | ∼ 90 | 10k | 18k | 157 | multiple |
| QV-highlights [23] | 845 | 10.2 | 16.1k | - | multiple |
| *Instructional Video Datasets* | | | | | |
| YouCook [10] | 2 | 88 | - | - | cooking |
| YouCook2 [48] | 176 | 2k | 13.8k | - | cooking |
| YouwikiHow [8] | - | 47k | - | 1398 | multiple |
| CrossTask [50] | 213 | 2.8k | 21k | 18 | multiple |
| COIN [41] | 476 | 11.8k | 46.4k | 180 | multiple |
| HT-Step Seen Val+Test Set [28] | 124 | 1.2k | 7k | 177 | cooking |
| HT-Step (Ours, Full) | **2.1k** | **19.7k** | **116k** | **433** | cooking |
| *HT-Step splits* | | | | | |
| HT-Step Train | 1.9k | 17.4k | 103k | 401 | cooking |
| HT-Step Val seen | 64 | 600 | 3.4k | 120 | cooking |
| HT-Step Test seen (S1) | 61 | 600 | 3.6k | 120 | cooking |
| HT-Step Test unseen (S2) | 116 | 1000 | 5.7k | 32 | cooking |

research over the last decade [2, 6, 11, 13, 22, 34, 37, 47], largely driven by the release of benchmark datasets [22, 41, 50]. Early datasets such as Breakfast [22] and 50Salads [40] are manually recorded, small-scale (spanning at most 100 hours) and limited in activity diversity (e.g., covering only breakfast recipes [22] or IKEA furniture assembly [16]). More recent datasets, such as COIN [41] and CrossTask [50], have capitalized on public video collections, including videos sourced from YouTube, in order to substantially increase dataset scale and activity diversity. A key characteristic of these videos is that they are instructional, i.e., they typically involve an individual teaching a complex task by narrating the sequence of steps.

Our HT-Step dataset also provides temporal annotations for steps of instructional videos. However, it surpasses existing datasets in terms of scale and activity diversity, as the steps are sourced from real-world instructional articles (wikiHow [20]) rather than being manually defined. This approach leads to much richer step descriptions, including a paragraph providing detailed instructions. Despite its exclusive focus on the cooking domain, HT-Step is significantly larger in scale and step description complexity. Our work significantly expands the small-scale val/test set that was recently released [28] for evaluating step grounding approaches trained *without* annotated temporal segments. While the focus of that work was the design of weakly supervised methods leveraging narrations as "free" signal for step-to-video alignment, here we share the first large-scale dataset enabling strongly-supervised training of methods for temporal article grounding. Our annotations over a large-scale training set, and a new test set of unseen tasks (not included in the training set) provide almost 20× more annotated videos and number of segments. We also include additional metadata per step (indicating *full* or *partial* segment-step matches). Furthermore, we include a comprehensive experimental evaluation of strongly supervised baselines on this task.

**Instructional Video Understanding** There exist various tasks related to instructional video understanding, such as step classification [25], step detection [41, 46], action segmentation [13], captioning [49], video retrieval [15], visual object grounding [29], and temporal grounding [26]. In this paper, we focus on the recently-introduced task of temporal article grounding, which has been tackled only with weakly-supervised approaches so far [8, 28]. In particular Chen et al. [8] is the first work to train and evaluate models for the temporal article grounding task. However this work was limited to weakly-supervised training due to the lack of a strongly labeled dataset, and used CrossTask as a proxy dataset for evaluation, with noisy mapping between article steps and CrossTask labels. In contrast, we introduce a large amount of strong temporal segment annotations

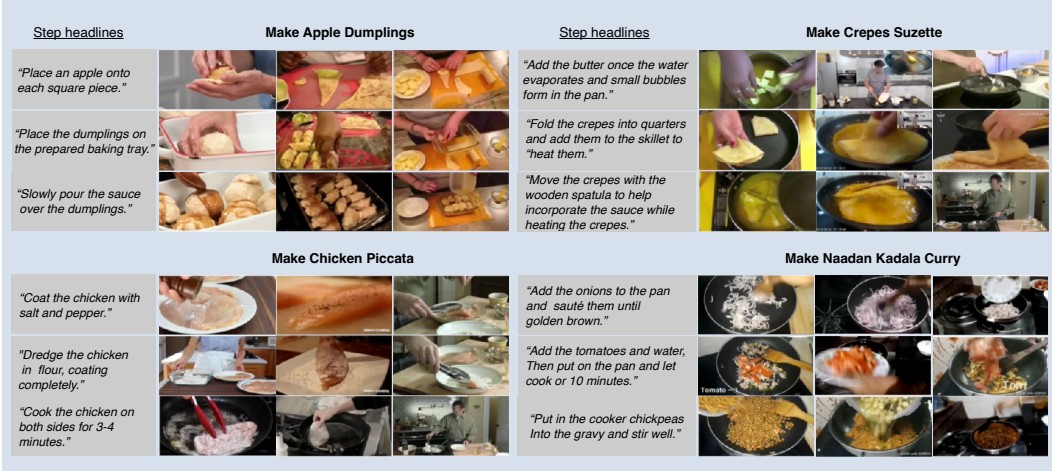

Figure 2: Sample frames for 12 videos spanning 4 tasks from the HT-Step training set. For each video, we show relevant frames for three steps. By annotating multiple videos of the same task, we obtain multiple instances of each step, which have high variations in appearance and viewpoints.

for training, and propose a metric that evaluates both the recall and precision of models, and is thus better suited to the task of temporal article grounding which involves multiple ungroundable steps. The temporal article grounding task is closely related to two other grounding tasks: (a) temporal grounding of a single natural language query [14, 23] (also known as moment retrieval) and (b) video paragraph grounding [4]. Benchmarks [14, 21, 23] for these tasks typically consist of video captions or questions that are loosely related to each other and which were obtained by human annotators based on the video contents. Thus, most approaches assume that each of the queries is groundable in the video. For example, video paragraph grounding aims at predicting a single temporal segment for each sentence of the paragraph. Instead, in our HT-Step benchmark the step queries originate from wikiHow articles, which might contain steps that are not shown in the video (non-groundable steps) or steps that are partially executed (partial matches). This makes it challenging to adapt temporal grounding approaches for temporal article grounding in our benchmark.

# 3  HT-Step

The creation of HT-Step involved pairing videos from HowTo100M with instructional articles from wikiHow and then annotating segments in each video with steps of the associated article. wikiHow[2] is an online knowledge base that houses thousands of articles with step-by-step instructions for a large variety of activities. Every article is typically structured as a series of steps, each step anchored by a descriptive headline and further supplemented with a detailed paragraph. HowTo100M, on the other hand, contains an extensive compilation of instructional videos that were collected using titles of wikiHow articles as search keywords for a subset of activities deemed to contain visual demonstrations. As a result of this sourcing procedure, each video in HowTo100M comes with a task label that corresponds to a specific wikiHow article. These pairings create a natural opportunity for annotating the HowTo100M videos with the step labels of the corresponding wikiHow articles, which we exploited for creating HT-Step. Next we describe in detail our annotation process.

## 3.1  Annotation workflow

**Activity selection.** HowTo100M spans a variety of domains, including cooking, DIY, crafts, sports, and gardening. As in previous works [17], we chose to focus on cooking activities as i) they account for approximately a third of the HowTo100M videos, and ii) they are relatively low-complexity tasks which can be annotated by non-experts. Using metadata, we selected 495 cooking activities (detailed list in the Appendix) for which HowTo100M contains at least 70 videos.

**Step Taxonomy.** We automatically sourced the step descriptions for all the activities from wikiHow. For every activity, we extracted the steps from the corresponding wikiHow article, forming a step taxonomy that includes 4, 958 unique steps. We list the full taxonomy in the Appendix.

---

[2]https://www.wikihow.com/

**Video validation.** For every video, the annotators were given the corresponding activity title and step taxonomy. They were then asked to watch the whole video and determine whether it contains the specified activity. If the video was found not matching the task, it was rejected.

**Temporal annotation.** During the temporal annotation loop annotators were asked to provide temporal boundaries for the wikiHow steps represented in all the non-rejected videos. An example annotation is illustrated in Figure 1.

**Task variations.** wikiHow articles often list several variations for a given task. For example, the instructional recipe for *"Make Taco Salad"* includes three versions: basic, deluxe and vegetarian. For such activities, we asked annotators to choose the variation that best fits the video and to annotate only the steps listed in that variation.

**Full vs partial matches.** We asked the annotators to mark a step as a "full" match if (i) all the instructions and ingredients listed in the step headline are shown in the segment and (ii) the step is brought to full completion. Conversely, "partial" indicates that some sub-steps in the step headline are skipped, some ingredients are not used, or that a step is interrupted and continued later.

**QA process.** To ensure the quality of the annotations, we followed a rigorous multi-stage Quality Assurance (QA) process. In the first stage, the videos were annotated by a single annotator. These initial annotations were then reviewed by more experienced annotators, who either approved all the annotations on a given video (all the marked steps were correct and no steps were missing) or marked it for re-doing, with specific comments indicating the necessary correction. At the last stage of the QA process, the annotations that were marked as incorrect were redone by a third set of annotators. Due to budget constraints, the full QA was enforced on a sample of roughly $13\%$ of the total annotations.

**Annotation output.** The annotation process took approximately 33k person-hours by 90 professional human annotators. Overall, 34% of the videos were rejected as not matching the specified activity, resulting in annotations for $433$ of the initial $494$ activities. We summarize the statistics of the resulting annotations and compare to existing datasets in Table 1.

## 3.2 Training, validation, and test splits

We formed two types of test sets: one for *seen activities*, i.e., activities that are also included in the training set, and one for *unseen activities*, i.e., novel activities not appearing in the training set. For seen activities we also formed a validation (val) set containing the same activities.

**Seen val/test set (S1).** We directly adopt the HT-Step validation and test splits [28] as the seen evaluation sets of our dataset, each containing 600 videos, 5 samples for each of 120 activities.

**Unseen test set (S2).** To create the unseen test set, we selected 32 activities that do not appear in the training set or the seen val/test set. We include all the annotations on videos of these activities that went through the QA review, resulting in a test set of 1000 annotated videos. This set is imbalanced w.r.t. activities, with the video count per activity ranging from 17 to 70.

**Training set.** All the other annotated videos are included in the final training set, which contains more than 17k videos and 100k annotated segments. We summarize the split statistics in the bottom part of Table 1.

## 3.3 Properties

**Article steps.** As previously mentioned, step descriptions in wikiHow articles have two parts: a headline and a paragraph. The headline is brief and may not reveal enough information for grounding; the paragraph contains additional details about the execution of the step. Annotators were asked to localize each step by considering both. This allowed us to obtain annotations where the paragraph description is essential for grounding the steps. We show some examples in Figure 3.

**Composite steps.** Many wikiHow steps are composite, i.e., include multiple sub-steps. Two examples of such steps are shown in Figure 4 (b,c). We introduced the labeling of partial vs full matches (discussed in Section 3.1) to account for the fact that composite steps are rarely fully represented in videos or they may be carried out in disjoint time intervals of the video.

**Non-groundable steps.** Some of the steps listed in the wikiHow article may not be represented at all in the video. An example is illustrated in Figure 1. Temporal grounding models that are trained/evaluated on this data must support handling queries with no associated temporal segments.

## 3.4 Statistics

In this section we provide insightful statistics about the dataset. See Appendix for more.

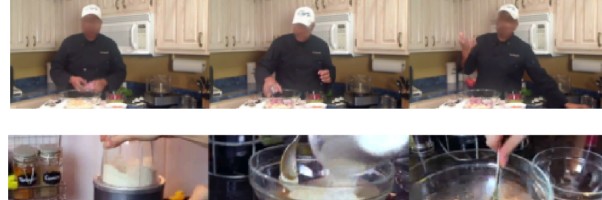

**Dice the celery, carrot, and pickle.** Use a sharp knife to carefully dice enough red onion to measure 1/4 cup (40 g). You'll also need to dice enough celery to measure 1/4 cup (55 g) and enough pickle to measure 1/4 cup (35 g). *Transfer the vegetables to the bowl with the mashed chickpeas.*

**Make the cake ball dough.** Place two-inch (5-cm) cubes of cake into the food processor and *pulse until the cake is broken up into fine crumbs. Transfer the crumbs to a mixing bowl.* [...] If you don't have a food processor, place the chunks of cake into a mixing bowl and mash them with a fork or break them up with your hands.

Figure 3: Qualitative examples where the paragraph is essential to localize the step. The paragraph sentences that are visually demonstrated in the video are highlighted in yellow.

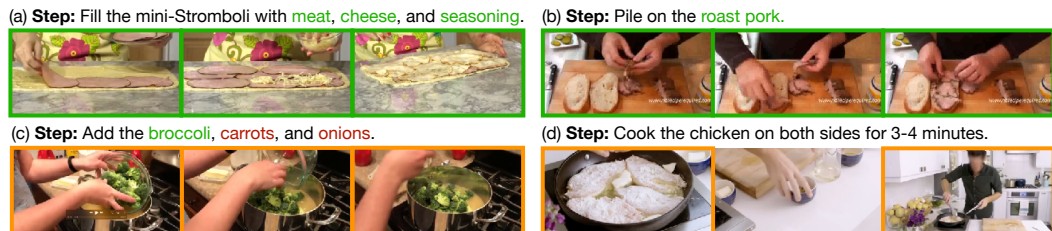

(a) **Step:** Fill the mini-Stromboli with meat, cheese, and seasoning.

(b) **Step:** Pile on the roast pork.

(c) **Step:** Add the broccoli, carrots, and onions.

(d) **Step:** Cook the chicken on both sides for 3-4 minutes.

Figure 4: **Examples of composite steps and full/partial matches.** Frames annotated as "full matches" are shown in green, "partial matches" are shown in orange. **(a)** An atomic step, fully matched. **(b)** A composite step: it is labeled as a full match as all the ingredients and the substeps mentioned in the paragraph are shown in the video. **(c)** Another composite step: here not all the described ingredients are utilized and thus it is labeled as a "partial match." **(d)** An example of step fragmentation: the step is executed over two disjoint segments which are consequently labeled as "partial matches."

**Overview.** Table 1 summarizes the statistics of HT-Step. It is larger and has a much richer step taxonomy than existing datasets. Compared to COIN, it has $4.8\times$ the duration, $2.6\times$ the number of annotated segments, $7\times$ the taxonomy size (4958 vs 778 unique steps), and $2\times$ the number of videos.

**Segment duration.** According to Figure 5a, the average segment duration is approx. 15 seconds, with most of the annotated segments at 10 seconds or shorter and a long tail of rare longer segments.

**Video coverage.** We define video coverage as the percentage of a video that is covered by annotations. Figure 5b shows the distribution of the average video coverage per activity, with a mean of 22%.

**Step coverage.** Step coverage denotes the fraction of unique steps from the associated wikiHow article that are annotated in a video and is on average 42%. Figure 5c shows that even for the activities with the highest step coverage, many article steps have no relevant temporal segments in each video.

**Distribution of segments over activities.** Figure 5d shows how the number of annotations is distributed over the $4,958$ steps. On average we get 24.6 segments per step. Since the choice of steps to be annotated was determined by the data, several steps have very few training samples, as indicated by the large concentration in the first bucket. We believe that this property will encourage the development of methods that can perform zero- or few-shot detection, as well as techniques that approach the task as language-based temporal grounding as opposed to class-based detection.

**Length of text descriptions.** We calculate the number of words as a proxy for the complexity of the step descriptions. In Figure 5e we provide the distribution of this metric averaged over activities, for HT-Step and COIN [41]. It is clear that HT-Step contains richer text descriptions, even when only the headlines are used (8.5 words vs 4.9 for COIN on average). When paragraph text is included in this calculation, the word count increases even further, to 17.1 words on average.

**Step ordering variation.** The steps in a wikiHow article are listed according to a recommended order of execution. In order to capture real-world deviations from these canonical orderings, we measure the Normalized Edit Distance (NED) between the annotated step sequence and the wikiHow listing, normalized over the length of the annotated sequence in each video. We calculate the mA-NED as the mean of the average NED per activity. We compute the same metric for the COIN dataset, using the ordering provided with the taxonomy of this dataset. Steps in our dataset are much more frequently executed out of order compared to COIN (mA-NED is 0.37 vs 0.27), as shown in Figure 5f. This presents a much more challenging temporal grounding setting.

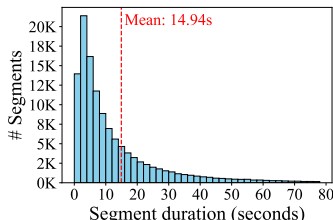

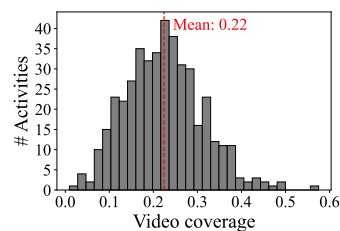

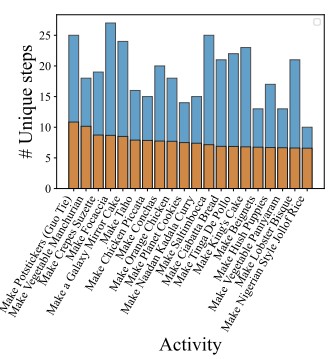

(a) **Distribution of segment durations.** The average segment duration is about 15 seconds, with most of the annotated segments at 10 seconds or shorter, and a long tail of rare longer segments.

(b) **Distribution of video coverage, averaged per activity.** Video coverage denotes the percentage of a video that is covered by step annotations. It exhibits a normal distribution around the mean.

(c) **Average number of unique steps** annotated per video for the 20 activities with most unique steps.

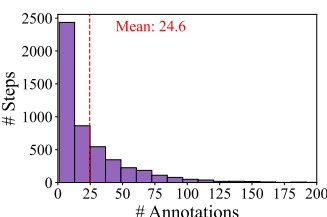

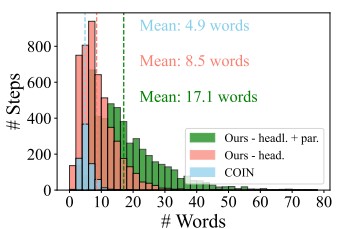

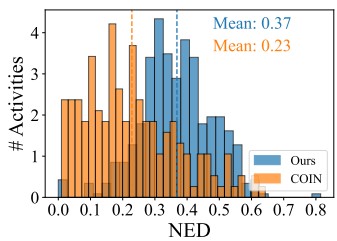

(d) **Distribution of annotations.** Annotations follow a long-tail distribution which should encourage the development of zero/few-shot detection or text grounding methods.

(e) **Comparison of text description lengths.** HT-Step contains long step descriptions, almost twice as long as those of COIN, even when only the headline text is considered.

(f) **NED.** Step sequences in HT-Step have a 0.37 NED compared to the canonical step ordering on average, which makes it more challenging compared to previous benchmarks.

Figure 5: **HT-Step statistics**. We provide various statistics for a qualitative overview of the dataset.

**Full vs partial matches.** Of the resulting annotated segments, 27% are full and 73% are partial matches. Some examples of full and partial match annotations are shown in Figure 4.

**Variations.** Most tasks (335 out of 433) involve a single listing of steps in wikiHow. However there is a significant number of activities (98) that was annotated according to more than one (and up to 6) sequences of steps. Refer to Appendix for an overview of the variations distribution.

# 4  Benchmark & Experiments

Table 2: **Article grounding mAP.** Comparison of state-of-the-art approaches on our benchmark. TS denotes the TimeSformer backbone.

| Method | Training data | Video Backbone | Text Backbone | Seen (S1) ↑ mAP@IOU @0.3 | @0.5 | @0.7 | @[0.3-0.7] | Unseen (S2) ↑ mAP@IOU @0.3 | @0.5 | @0.7 | @[0.3-0.7] |
|---|---|---|---|---|---|---|---|---|---|---|---|
| *Detection* | | | | | | | | | | | |
| ActionFormer [46] | HT-Step | TS [5, 25] | - | 32.9 | 23.6 | 12.6 | 23.1 | - | - | - | - |
| ActionFormer [46] | HT-Step | S3D [30, 44] | - | 35.1 | 25.6 | 14.8 | 25.4 | - | - | - | - |
| *Grounding* | | | | | | | | | | | |
| UMT [26] | HT-Step | S3D [30, 44] | Word2Vec [32] | 8.0 | 4.0 | 1.5 | 4.4 | 4.9 | 2.3 | 0.8 | 2.6 |
| UMT [26] | HT-Step | TS [5, 25] | CLIP [33] | 15.7 | 8.7 | 3.2 | 9.1 | 9.4 | 4.9 | 1.7 | 5.3 |
| MT+BCE | HT-Step | S3D [30, 44] | Word2Vec [32] | 31.5 | 19.6 | 8.1 | 19.7 | 18.6 | 10.3 | 3.7 | 10.6 |
| MT+BCE | HT-Step | TS [5, 25] | Word2Vec [32] | 26.2 | 16.5 | 6.0 | 16.1 | 13.5 | 6.8 | 2.5 | 7.4 |
| ActionFormer-T | HT-Step | S3D [30, 44] | Word2Vec [32] | 27.8 | 20.0 | 10.6 | 19.6 | 10.8 | 6.4 | 3.1 | 6.6 |
| ActionFormer-T | HT-Step | S3D [30, 44] | MPNet [39] | **36.9** | **26.5** | **15.3** | **26.3** | **27.4** | **18.3** | **9.3** | **18.4** |
| VINA [28] | HT100M | S3D [30, 44] | Word2Vec [32] | 12.6 | 4.7 | 1.2 | 5.9 | 8.2 | 3.1 | 0.6 | 3.8 |
| MT+BCE (VINA) | HT100M+HT-Step | S3D [30, 44] | Word2Vec [32] | **46.2** | 29.9 | 12.9 | 29.8 | **31.6** | 18.7 | 7.7 | 19.3 |
| ActionFormer-T | HT100M+HT-Step | S3D [30, 44] | MPNet [39] | 41.2 | **30.8** | **18.3** | **30.2** | 29.7 | **20.3** | **10.7** | **20.4** |

**Task definition.** The annotated data of HT-Step can be used to train and evaluate models for a plethora of tasks, from temporal article grounding to task-graph-based step localization [47], narration-based step grounding [28], captioning[21], and step forecasting [25]. In this paper, we focus on the temporal article grounding task: given an untrimmed video and an instructional article containing a sequence of steps, a model must predict temporal segments for the steps demonstrated in the video, along with a confidence score for every segment.

**Evaluation protocol.** For evaluating models, we introduce the *article-grounding mAP* – a variant of the popular mean Average Precision (mAP) metric from object detection [12, 24]. AP is separately computed *for each activity* by treating all steps within the activity as class-agnostic text queries. The per-activity APs are then averaged to obtain the article-grounding mAP. Following prior work [18], we evaluate AP under different temporal Intersection-over-Union (tIoU) thresholds.

## 4.1   Results

We evaluate state-of-the-art approaches and baselines from step detection, single-step grounding, and multi-step grounding on our task. Full architecture, training, and implementation details can be found in Appendix.

**ActionFormer** [46] is a state-of-the-art, multi-scale action detection model that operates on a fixed taxonomy of steps (i.e., it does not use language), and as a result, can only detect seen steps (S1 split).

**ActionFormer-T** is a variant of the ActionFormer model which we adapted to enable open-vocabulary step detection (*i.e.* unseen step grounding). It is constructed by replacing the classification head with a dot product between the textual embeddings of every step and the visual embeddings obtained from the temporal feature pyramid.

**UMT** [26] is a transformer-based model built for joint moment retrieval and highlight detection, and represents the state-of-the-art in single-step grounding.

**VINA** [28] is a model for temporal article grounding. Unlike traditional single-step grounding models, it can capture step ordering and the relationship between steps, and was trained with weak supervision from ASR transcripts and video-article pairs.

**MT+BCE** is our modification of VINA for fully-supervised temporal article grounding. It embeds the full *sequence of step headlines* along with the video clips, and feeds them together to a multimodal transformer. We train it with a per-frame binary cross entropy loss (BCE) and produce segment predictions using a 1D blob detection routine [43].

**Baselines comparison.** Table 2 compares the performance of the aforementioned models for different choices of visual and textual backbones on the seen and unseen article test splits. Among all models trained from scratch on our training set, ActionFormer-T achieves the best performance on the seen test set (S1), while the performance of ActionFormer is slightly lower but comparable (26.3% vs 25.4% mAP). We conjecture that the structure of our training set, which has multiple video examples per activity, allows the ActionFormer to learn step representations without explicitly requiring language descriptions. However, modelling the language explicitly with ActionFormer-T yields a small boost, as it enables capturing similarities and subtle differences between step descriptions. More importantly, ActionFormer cannot be applied to detect steps of novel activities and articles in the unseen test set (S2). On this test set, our vanilla MT+BCE baseline that leverages the article structure (in contrast to grounding each step independently) performs reasonably well, outperforming UMT, even though the latter has been trained with a regression loss for more accurate temporal boundaries. This shows that leveraging the article step structure is very helpful for grounding novel articles in video. However, although ActionFormer-T has the disadvantage of performing single-step grounding, it outperforms all models in the unseen split too, presumably because it combines an architecture tailored to multi-scale detection with language modelling.

**Weakly supervised pretraining.** The state-of-the-art weakly supervised VINA model trained on HowTo100M ASR captions and wikiHow articles yields a low mAP metric since it was not trained with accurate temporal boundaries. However, using the same pretrained VINA model as initialization to finetune our MT+BCE baseline on HT-Step leads to a remarkable performance boost, outperforming all previous baselines by a large margin, achieving 29.8 and 19.3 mAP on S1 and S2 respectively. We perform a similar experiment by pre-training the ActionFormer-T (AF-T) model on the ASR captions of HowTo100M, then fine-tuning the model with the HT-Step annotations. The results are similar, with the model achieving very good performance in the high IOU evaluation settings, leading to the best overall results, *i.e.* 30.2 and 20.4 mAP on S1 and S2 respectively. The VINA-pretrained model still shows better performance on the low-IOU evaluation, reflecting the strong localization abilities of the original model. These experiments clearly showcase the great

potential in combining strong step labels from HT-Step with noisy ASR captions from HowTo100M for training grounding models.

Table 3: **Ablation of text used for step description.**

| Method | Text backbone | Text input | | | ↑ mAP @ [0.3-0.7] | |
|---|---|---|---|---|---|---|
| | | Hdl. | Prg. | Act. | seen (S1) | unseen (S2) |
| ActionFormer-T | MPNet | ✓ | ✗ | ✗ | 26.3 | 18.4 |
| ActionFormer-T | MPNet | ✓ | ✓ | ✗ | 26.1 | 18.0 |
| ActionFormer-T | MPNet | ✓ | ✓ | ✓ | 24.2 | 14.7 |
| ActionFormer-T | MPNet | ✓ | ✗ | ✓ | 24.0 | 13.7 |
| ActionFormer-T | MPNet | ✗ | ✓ | ✗ | 22.8 | 13.9 |
| ActionFormer-T | MPNet | ✗ | ✓ | ✓ | 21.1 | 11.3 |
| MT + BCE | Word2Vec | ✓ | ✗ | ✗ | 19.4 | 10.1 |
| MT + BCE | Word2Vec | ✓ | ✓ | ✗ | 19.2 | 10.3 |
| MT + BCE | Word2Vec | ✓ | ✗ | ✓ | 19.7 | 10.6 |
| MT + BCE | Word2Vec | ✗ | ✓ | ✗ | 17.0 | 8.3 |
| MT + BCE | Word2Vec | ✓ | ✓ | ✓ | 19.8 | 10.1 |

Table 4: **Partial vs full matches**.

| Method | seen | | unseen | |
|---|---|---|---|---|
| | partial | full | partial | full |
| *Detection* | | | | |
| ActionFormer [46] | 20.4 | 29.6 | - | - |
| *Grounding* | | | | |
| MT + BCE [17] | 16.5 | 24.7 | 8.8 | 8.7 |

**What text information is important for article grounding?** We experiment with different combinations of a step's headline, paragraph and activity name to form its description and summarize the results in Table 3 (details for how every model combines text sources are provided in Appendix). It is evident that the headline contains the most valuable information. Paragraph information alone is enough for learning the task, however performance when using only paragraphs is lower than with headlines. Results from incorporating the activity name in the step descriptions vary depending on the model. Overall, although paragraph information is essential for accurate grounding, as shown in Figure 3, combining headline and paragraph provides marginal improvements. In addition, with all the approaches, there is a significant performance gap between the seen and the unseen test set. We conclude that temporal article grounding on our benchmark presents a challenging and exciting problem that will hopefully inspire the development of better models by the community.

**What are the results on partial matches compared to full matches?** To understand the differences between partial and full matches, we take models trained on the complete set and test them on subsets containing only partial-match or only full-match labels (details in Appendix). We present the results in Table 4. When evaluating on seen activities (S1), the performance of all models is significantly better on the full-match subset. This aligns with our intuition that full-match annotations should be easier to ground, as their step descriptions align better to what is visually demonstrated in the video. However, this gap disappears when evaluating on unseen activities (S2). We conjecture this may be due to full step matches being visually more consistent, resulting in models overfitting them on seen activities.

Table 5: **Comparison to SOTA for zero-shot step grounding on the CrossTask dataset.**

| Method | ↑Avg. R@1 (%) |
|---|---|
| Zhukov [50] | 22.4 |
| HT100M [31] | 33.6 |
| VideoCLIP [45] | 33.9 |
| MCN [7] | 35.1 |
| DWSA [38] | 35.3 |
| MIL-NCE [30] | 40.5 |
| VT-TWINS [19] | 40.7 |
| UniVL [27] | 42.0 |
| AF-T (HT100M) | 37.1 |
| AF-T (HT100M + HT-Step) | **48.5** |

Table 6: **Zero-shot article grounding evaluation on CrossTask.**

| Training data | ↑Avg. R@1 (%) | mAP@[0.3-0.7] |
|---|---|---|
| HT100M | 37.1 | 5.2 |
| HT100M + COIN | 41.4 | 6.7 |
| HT100M + HT-Step | **48.5** | **9.5** |

**Do models trained on HT-Step generalize to other tasks?** We showcase the potential of leveraging HT-Step for model pre-training by evaluating one of our models on zero-shot atomic step localization. In particular, we pre-trained the ActionFormer-T (AF-T) model on the ASR captions of HowTo100M (following prior work [27, 30, 45]), then fine-tuned the model with the HT-Step annotations. We perform evaluation on the CrossTask dataset, following standard protocol [50], i.e., we compute the recall per task based on a single predicted timestamp per step, computed over 20 random subsets of the training videos. As Table 5 demonstrates, it is clear that training on HT-Step results in a

huge boost in performance, namely a +11.4% absolute improvement, but also surpasses the previous state-of-the-art on this benchmark by a large margin (+6.5%).

To further motivate the usefulness of the HT-Step annotations compared to existing datasets, we train the same model on the COIN dataset [41], and perform zero-shot evaluation on CrossTask, using both the standard recall metric (for single timestamp prediction evaluation) as well as mAP (for temporal segment and precision evaluation). The results are shown in Table 6. It is clear that fine-tuning on the HT-step annotations results in a much more substantial performance improvement compared to fine-tuning on COIN (+7.1% R@1 and +2.8 mAP).

## 5 Conclusion

We have introduced HT-Step, a dataset containing temporal annotations of instructional article steps in cooking videos. The dataset offers a large-scale training resource for the task of temporal article grounding, presenting new challenges and encouraging the development of alignment methods that leverage order and procedural structure. By releasing the annotations and providing benchmark protocols, we aim to spur new research in this domain and advance the field of procedural activity understanding.

**Limitations and societal impact**  We acknowledge that HT-Step is intended for research purposes and should not be regarded as a comprehensive dataset encompassing the full range of human activities. Models trained on our dataset may exhibit biases towards the specific activities included in the dataset, resulting in a limited coverage of our everyday living scenarios.

**Acknowledgements.**  We thank Mandy Toh, Yale Song, Gene Byrne, Fu-Jen Chu, Austin Miller, and Jiabo Hu for helpful discussions and invaluable engineering support.

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

# Appendix

This Appendix provides more comprehensive statistics about the dataset (Section A), architecture and implementation details for the baselines discussed (Section B), a breakdown of model performance by activity (Section C), and extra details about the evaluation protocol (Section D).

## A    Extra statistics

Figures 6a-6i contain additional statistics about HT-Step.

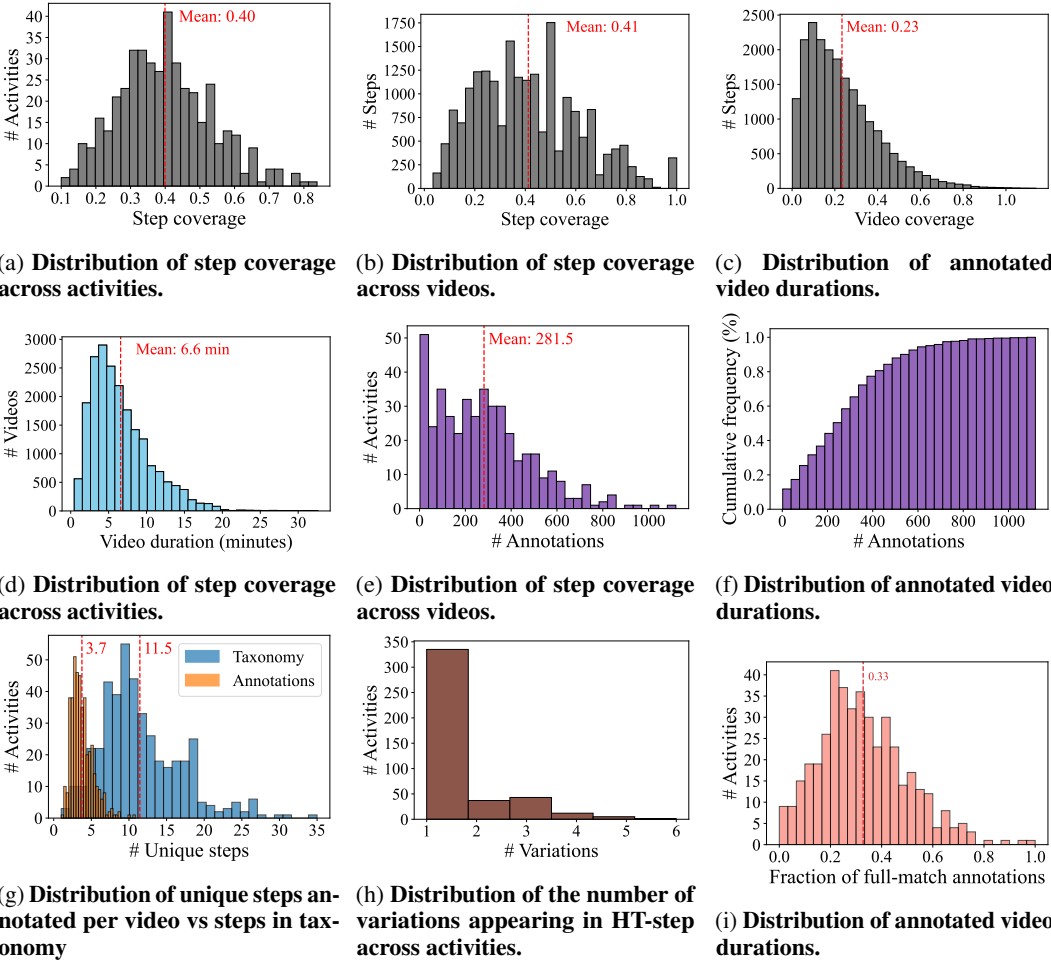

(a) **Distribution of step coverage across activities.**

(b) **Distribution of step coverage across videos.**

(c) **Distribution of annotated video durations.**

(d) **Distribution of step coverage across activities.**

(e) **Distribution of step coverage across videos.**

(f) **Distribution of annotated video durations.**

(g) **Distribution of unique steps annotated per video vs steps in taxonomy**

(h) **Distribution of the number of variations appearing in HT-step across activities.**

(i) **Distribution of annotated video durations.**

Figure 6: Additional statistics of HT-Step.

## B    Baselines implementation details

We train all the baselines on top of the same feature sequences, extracted from frozen backbones.

**TimeSformer (TS) features.**    The TimeSformer features are extracted using the public model[3] of [25] pre-trained with distant supervision on HowTo100M. We obtain 1 feature per second, by resampling the video at 8 fps and extracting features with a stride of 8 frames. The feature dimensionality is 768.

---

[3] https://dl.fbaipublicfiles.com/video-distant-supervision/TimeSformer_divST_8x32_224_HowTo100M_pretrained.pth

**S3D features.** The S3D features are extracted using the published model[4] of [30] pre-trained with MIL-NCE [30]. We obtain 1 feature vector per second, by resampling the video at 16 fps and extracting features with a stride of 16 frames. The feature dimensionality is 1024.

**ActionFormer.** We use the official ActionFormer implementation[5] provided by the authors[46]. We set the number of classes to 4958 *i.e.*, one detection output for each step in the taxonomy. We set the max sequence length to 512 and train for 20 epochs with a batch size of 16, using the AdamW optimiser with cosine learning rate schedule, base learning rate of $1e-4$ and 5 warm-up epochs on a single Nvidia Tesla V100 GPU with 32GB of memory.

**ActionFormer-T.** ActionFormer-T is trained with the exact same hyper-parameters, loss and labels as ActionFormer. We extract the text representations, using the MPNet implementation provided by the `sentence_transformers` library[6]. The text embeddings are frozen, so no gradients are backpropagated into the pre-trained langauge model. We use a 2-layer MLP with hidden side 512, ReLU activation and Layer Normalization, to project and transform the $768-$dimensional text embeddings into the $512-$dimensional video embedding space. Step text descriptions from different sources (e.g. headline, paragraph, activity, see also Table 3 of the main paper) are combined by simple concatenation at the text level, *i.e.* the combined sentences including all three have the form "Activity: Headline. Paragraph".

**UMT.** For UMT, we use the authors' official code [7]. We train models with learning rate $1e-3$ and batch size 64 and train for 200 epochs. We use only the unimodal encoder for video (we do not use the audio encoder, or the cross-attention modules). All remaining hyperparameters follow the configuration for the QVHighlights task provided by the authors.

**MT+BCE.** The input to our temporal article grounding baseline is a temporal sequence of visual features extracted with a sliding window (using either the TimeSformer or S3D backbones as explained above), and a sequence of step sentences (consisting of the activity name and the article step headlines). We base our model on the VINA [28] architecture by removing the additional narrations modality, i.e., we do not use the narration unimodal encoders, positional encodings and the alignments of steps to narrations or narrations to video. We use the TAN[8] codebase for our implementation. All of the architecture hyperparameters (e.g., number of Multimodal Transformer layers, embedding dimensions etc.) are adopted from VINA. The only difference is the maximum length of the input video which we increase to 1200 seconds to account for the longer videos in the HT-Step training set.

To obtain temporal segment predictions for each article step from the Multimodal Transformer outputs, we: (1) compute the normalized dot product between each step contextual embedding and each video clip contextual embedding. This results in a $T \times S$ alignment matrix, where $T$ is the number of timesteps and $S$ is the number of steps. (2) We pass these similarities through a sigmoid activation (with temperature 0.07) to obtain a confidence score about whether each timestep $t$ is aligned with step $s$, (3) we post-process the temporal sequence of confidence scores for each step with an 1D blob detection routine to obtain temporal segments at multiple scales. In particular, we apply Laplacian of Gaussian filters at 13 scales, covering Gaussian standard deviations from 1 to 480 [43].

The model is trained with binary cross-entropy loss applied at each temporal timestep and for each article step. We train our model for 9 epochs using the same optimizer, learning rate and batch size as VINA [28].

Adding paragraph information: For our ablations in Table 3 of the main paper, we added paragraph information to the MT+BCE model simply by interleaving step headlines with step paragraph sentences. In other words, we tokenize the article into sentences (with a maximum of 28 sentences per step) and we encode and feed that sequence of sentences to the Multimodal Transformer. We use the same positional encoding for all sentences associated with the same article step. In order to

---

[4] https://github.com/antoine77340/S3D_HowTo100M
[5] https://github.com/happyharrycn/actionformer_release
[6] https://huggingface.co/sentence-transformers/paraphrase-mpnet-base-v2
[7] https://github.com/TencentARC/UMT/tree/main
[8] https://github.com/TengdaHan/TemporalAlignNet

obtain a single contextual embedding for each step, we max pool the embeddings of the headline and paragraph sentences of that step.

Model weights initialization: We train all variants of MT+BCE from scratch, except for the model in the last row of Table 2, which was trained after initializing the unimodal encoders, positional encodings and Multimodal Transformer weights using a VINA model pretrained on the HTM370k [17] subset of HowTo100M using pseudo-labels for wikiHow steps (and no ASR narrations) [28]. For this experiment, we adopt the same maximum video length as VINA (1024 seconds).

## C    Per-activity predictions

In Table 7 we show the per-activity AP breakdown of the performance of the two best models. We show the 25 highest and 25 lowest scoring activities, ranked by the performance of the ActionFormer detection model. Note that activities that are challenging for the fixed taxonomy, detection model (such as Cook Pork Tenderloin for which the AP is 0.62%) are handled better by the temporal grounding model (achieving 28.9% AP for Cook Pork Tenderloin). For this particular example of *Cook Pork Tenderloin*, this can be explained since this activity has only 4 examples in the training set. Therefore, the detection model does not have enough training samples to learn a good representation for the steps of this activity. On the other hand, the temporal article grounding model, that has been initialized with a model trained with weak-supervision on a much larger dataset (HTM370k) can perform better in this few-shot scenario. Another interesting observation is that for some activities the detection-based model outperforms language-based grounding.

## D    Evaluation protocol details

### D.1    Article-grounding AP metric

Approaches in our proposed temporal article grounding benchmark are evaluated using Article Grounding mean Average Precision (AGrd. mAP) over temporal IoU thresholds from 0.3 to 0.7 with a step size set to 1 (as in existing benchmarks [18]), and using three fixed tIoU thresholds at 0.3, 0.5 and 0.7. As explained in the main paper, our proposed metric computes an AP per activity (which might be associated with multiple articles if is has variations) by treating all article steps associated with that activity as class-agnostic text queries (similar to the temporal grounding Average Precision introduced in [23]). The per-activity AP is only computed on videos demonstrating each particular activity. The final article-grounding mAP is computed by averaging the per-activity APs. Our mAP-based metric is more suitable for the temporal article grounding task than existing recall-based metrics for grounding [8, 50] which ignore non-groundable steps, or frame-wise metrics for step detection [41], which ignore the temporal extent of each segment.

### D.2    Breakdown of article-grounding mAP per match type (*full* vs *partial*)

In Table 4 of our main paper, we report article-grounding mAP computed per step match type (*full* vs *partial*). The mAP for full matches was computed separately for step queries that only have fully-matching temporal segments (or no matching segments) in their corresponding video. Step queries that have both full and partial matches in the same video were ignored from the computation of the mAP on full matches. Furthermore, APs are only computed for activities that have ground-truth step queries with full matches and averaged over those. Overall, the mAP for full matches was computed based on 78 activities, with 1176 ground-truth instances. The mAP for partial matches was computed in a corresponding manner, covering 79 activities and 2375 ground-truth instances.

## E    Training, validation, and test splits

We have included details about the training, validation and test splits in Section 3.2 of the main paper. Here we add some comments.

**Seen val/test set (S1).** We note that these sets are balanced, each containing 600 videos in total, 5 videos per each of 120 activities, with an overlap between validation and test amounting to 63 activities. Labels are released for the val set, while labels for the seen test set are withheld and a

Table 7: **Breakdown of AP performance per activity on the seen test set (S1).** We show the 25 highest and 25 lowest scoring activities, ranked by the performance of the ActionFormer model.

| | Model | |
| --- | --- | --- |
| | ActionFormer | MT+BCE(VINA) |
| **Activity** | | |
| Make Lunch Box Oatmeal Cookies | 55.02 | 66.65 |
| Make Chicken Liver Pate | 51.94 | 45.56 |
| Deep Fry a Turkey | 51.56 | 43.02 |
| Make Tomato Pie | 47.97 | 38.43 |
| Make Buttermilk Fried Chicken | 45.53 | 39.40 |
| Bake a Sweet Potato Pie | 43.62 | 25.70 |
| Make Scotch Eggs | 42.90 | 41.82 |
| Make Pecan Crusted Blackened Catfish | 42.37 | 26.54 |
| Make Vegetable Paniyaram | 42.30 | 39.91 |
| Cook Arepas | 42.16 | 43.75 |
| Prepare Mexican Chilaquiles | 41.15 | 40.09 |
| Make Chiles Rellenos | 40.82 | 35.48 |
| Make Beef Stroganoff | 40.59 | 27.03 |
| Clarify Butter | 40.38 | 39.71 |
| Make Toad in the Hole | 38.56 | 43.81 |
| Make Focaccia | 38.54 | 39.11 |
| Clean Flounder | 37.95 | 15.60 |
| Make Chicken Piccata | 37.77 | 44.94 |
| Brine, Truss, and Roast a Turkey | 36.15 | 49.03 |
| Grill Bacon | 35.89 | 55.26 |
| Make Eggplant Pasta Sauce | 35.31 | 30.93 |
| Make Mofongo | 35.05 | 43.49 |
| Make Saltimbocca | 34.84 | 34.21 |
| Cook Brussels Sprouts with Chestnuts | 34.52 | 42.86 |
| Make Beignets | 34.34 | 40.71 |
| . . . | | |
| Make White Chili | 15.79 | 11.41 |
| Make Fairy Cakes with Self Raising Flour | 15.51 | 36.88 |
| Make Chicken Cacciatore | 15.42 | 28.50 |
| Make Grilled Artichokes | 15.19 | 27.35 |
| Make Healthier Fish Sticks | 14.98 | 23.45 |
| Bake a Queen Elizabeth Cake | 14.91 | 27.57 |
| Make Coconut Rice | 14.82 | 31.91 |
| Make Hostess Twinkies | 14.54 | 24.67 |
| Cook Cube Steak | 13.20 | 34.79 |
| Make Bannock | 12.75 | 17.94 |
| Make Mango Chutney | 12.74 | 11.38 |
| Make Overnight Caramel Pecan Rolls | 12.47 | 18.48 |
| Make Vegan Ceviche | 11.96 | 4.85 |
| Cook Black Eyed Peas | 11.44 | 19.02 |
| Make a Cheese Crisp | 9.32 | 14.16 |
| Cook Bacon in the Microwave | 9.14 | 40.50 |
| Make Italian Ice | 9.01 | 19.73 |
| Make Quick and Easy Sausage Rolls | 8.27 | 16.67 |
| Braai Steak | 8.08 | 9.27 |
| Make a Hearty Stew | 7.73 | 31.03 |
| Make Mediterranean Vegetable Cheese Pie | 6.19 | 16.37 |
| Make Hungarian Goulash | 6.00 | 18.43 |
| Make Bacon Toffee | 3.92 | 8.15 |
| Make Blueberry Strudel | 1.96 | 9.29 |
| Cook Pork Tenderloin | 0.62 | 28.94 |
| **mAP** | **25.4** | **29.8** |

fair evaluation protocol on this set is supported via a test server that will be made available to the community.

**Unseen val/test set (S2).** Note that the headlines or paragraphs of some steps in the unseen val/test sets may be very similar to steps of the activities included in the training set, due to the compositionality of recipes. For example, the unseen activity of *Make Poutine* contains the step "Add the garlic and shallot" which is similar to steps such as "Add the garlic and cook for 30 seconds" from the seen activity *Make a Hearty Stew* and "add the garlic slices and cook for 1 minute." from *Make Tumbet*. Evaluation on the unseen test set will be made possible through the test server.

