# OpenReview forum: "HT-Step: Aligning Instructional Articles with How-To Videos"
_NeurIPS.cc/2023/Track/Datasets_and_Benchmarks — NeurIPS 2023 Datasets and Benchmarks Poster_

### Official Review · Reviewer_ssNp · 2023-06-26
**A new dataset called HT-step, including pair of video clip and corresponding step instruction, is proposed. It might be helpful for video-instruction-alignment related tasks.**

**Rating:** 6
**Confidence:** 4
**Correctness:** It seems the descriptions and claims …
**Clarity:** The presentations are clear, but can …

**Strengths:**

1. The scale of the proposed dataset is relatively large.

2. There are no such segment-instruction dataset for now. Thus, it is beneficial for the video & language research community.

**Additional Feedback:**

No additional comments.

**Documentation:**

Sufficient details are included.

**Ethics:**

No ethical concerns since video are already public before.

**Limitations:**

No obvious limitations.


**Opportunities For Improvement:**

1. The annotation process can be described more clearly by providing pipeline figures and more examples.

2. It seems that the number of partial matches is quite large. It might be more helpful if these annotations are revised to match the video content.


**Relation To Prior Work:**

Related works and discussion are sufficient.

**Summary And Contributions:**

The authors proposed a new dataset, called HT-step, for the task of video-instruction aligning. The HT-step dataset contains 116K segment-instruction pairs related to cooking. The numbers of videos, segments and activities are the largest among similar datasets. The annotation process, basic analysis about the dataset are provided.

---

> ### Author Response · Authors · 2023-08-22
> **Rebuttal**
>
> **Q1. The annotation process can be described more clearly by providing pipeline figures and more examples.**
>
> We have added a visualization of the annotation pipeline in the updated version of the paper (Figure 2), and will include the full details in the final version. We withheld a few of the full details of the process (e.g., the annotation tools) to preserve the anonymity of the submission. Our submission also includes: examples of full/partial matches (Fig. 5 in updated version), of steps (Fig. 3 in updated version) and annotated videos (in Supplementary).
>
> **Q2. It seems that the number of partial matches is quite large. It might be helpful if annotations are revised to match the video content.**
>
> The occurrence of a large number of partial matches, i.e. article steps whose textual description does not perfectly align with the visual contents of the video, is an inherent challenge of the article grounding task. Maintaining a common taxonomy among multiple videos instead of obtaining video-specific captions was one of the requirements and design choices for this task.
> This is in contrast to vanilla natural language description grounding/captioning datasets where textual descriptions completely match the video, or action detection datasets, where a temporal segment completely matches its label.
>
> However, as demonstrated by our experimental results (see briefly bellow, also reply to Reviewer rjWS, Q4 and updated manuscript lines 291-305 and Tables 5&6), HT-Step can be a rich resource for training general natural language localization models too.
>
> | Method                           | Avg. Recall |
> |----------------------------------|-------------|
> | UniVL   - SOTA                         | 42.0        |
> | ActionFormer-T (HowTo100M)       | 37.1        |
> | ActionFormer-T (HowTo100M + COIN)| 41.4        |
> | ActionFormer-T (HowTo100M + HT-Step)| **48.5**    |
>
>
> That said, we agree with the reviewer that revising the textual descriptions of the collected video segments could unlock new tasks, such as step captioning. The current collection of temporal segment annotation would largely facilitate such an effort, but it is outside the scope of this paper and we leave it to future work.

---

### Official Review · Reviewer_rjWS · 2023-07-06
**This paper constructs a dataset, HT-Step, by collecting step information from wikihow and aligning to instructional videos. The dataset includes temporal intervals of rich step descriptions, enabling the novel vision-language task, i.e. temporal article grounding. HT-Step facilitates the research towards procedure activity understanding at scale.**

**Rating:** 7
**Confidence:** 4
**Correctness:** Yes
**Clarity:** Yes

**Strengths:**

1. This paper leverages the rich how-to instructions from wikihow, i.e. Activity, headline, and paragraph to annotate the instructional videos. HowTo100m has been widely studied by the community, but the poor ASR quality has limited the research. Recent works such as TAN, DistantSup attempted to improve the dataset by either aligning video with ASR or retrieving the most similar step description regarding ASR. HT-Step takes a step further by providing more detailed step descriptions and temporal segments. This dataset outperforms most procedure activity datasets in terms of videos, number of segments, and activities.
2. On various video backbones (S3D, TimeSformer), text backbones (Word2Vec, MPNet) and task formulations (activity detection, single-sentence grounding, article grounding), the proposed benchmark dataset as well as challenging subsets (i.e. Unseen activities) are fairly evaluated.

**Additional Feedback:**

1. The scale of the dataset is one of the advantages compared with prior procedure activity understanding datasets. It would be interesting to validate the potential of HT-Step on pre-training. For example, pre-training the model on HT-step and evaluating (e.g. Finetuning or zero-shot testing) on downstream dataset (e.g. CrossTask).

**Documentation:**

Yes

**Limitations:**

Yes

**Opportunities For Improvement:**

1. In Table3, the authors found that the paragraph alone is enough for article grounding. This is true for Actionformer-T but the result is not listed for MT+BCE. The paragraph is supposed to contain more fine-grained descriptions of the activity than the headline. Could the authors explain why using the headline alone (line 1) outperforms using only the paragraph (line 4)?
2. [1] also proposed a benchmark on temporal article grounding. The authors are encouraged to state the main difference.
[1] Long Chen et al. Weakly-supervised temporal article grounding. EMNLP 2022

**Relation To Prior Work:**

Yes

**Summary And Contributions:**

This paper constructs a dataset, HT-Step, by collecting step information from wikihow and aligning to instructional videos. The dataset includes temporal intervals of rich step descriptions, enabling the novel vision-language task, i.e. temporal article grounding. HT-Step facilitates the research towards procedure activity understanding at scale.

---

> ### Author Response · Authors · 2023-08-22
> **Author Rebuttal**
>
> We thank the reviewer for the thorough review and comments.
>
> **Q1. Could the authors explain why using the headline alone (line 1) outperforms using only the paragraph (line 4)?**
>
> This is expected since:
> - All article steps have a headline that contains a summary of the step.
> Conversely, paragraphs are not available for all steps (21% of the steps have no paragraph).
> - The paragraph often only contains tips, suggestions and potential pitfalls rather than step descriptions.
> - Paragraph information is usually complementary to what is already included in the headline (although of course there are cases where only some paragraph sentences are groundable and not the headline, e.g. first example shown in Figure 4).
>
> Some illustrative examples from the **Make Kimchi Jjigae** wikiHow article:
> - Step 2:
>   - Headline: *Once the clay pot is heated, add two table spoons of vegetable oil.*
>   - Paragraph: None
> - Step 9:
>   - Headline: *Generously add chicken broth into the pot.*
>   - Paragraph: *Around 28 to 30 ounces should suffice.*
>   - Note: The paragraph here contains information about the quantity of the ingredients that is not crucial for grounding.
>
> **Q2. In Table3, the authors found that the paragraph alone is enough for article grounding. This is true for Actionformer-T but the result is not listed for MT+BCE.**
>
> The result was already listed for MT+BCE in Table 3, second to last row. Similarly to Actionformer-T the paragraph alone leads to a slightly lower performance compared to using the headline alone (19.6 vs 23.7 mAP for Actionformer-T, 17.0 vs 19.4 mAP for MT+BCE).
>
> **Q3.  Chen et al. 2022 also proposed a benchmark on temporal article grounding. The authors are encouraged to state the main difference.**
>
> The work of [WSAG, Chen et al, 2022] was limited to weakly-supervised training due to the lack of a strongly labeled dataset, and used CrossTask as a proxy dataset for evaluation, with noisy mapping between article steps and CrossTask labels. In more detail:
>
> 1. They train with weak supervision, i.e. only the video-level activity labels instead of temporal segments, resulting in models that perform rather poorly, e.g., worse than  the MIL-NCE model trained on ASR.
> 2. For evaluation they resort to using the CrossTask dataset, by employing a heuristic for mapping CrossTask labels (short sentences) to wikiHow articles. In fact, in the Limitation section of the paper, the authors acknowledge several issues with this: “When manually mapping the “step” in CrossTask to the “sentence” in the wikiHow article, we found it not always be one-to-one perfect mapping. In a few cases, multiple sentences may refer to a single step or multiple steps may refer to a single sentence. Thus, the original ground-truth annotations for each CrossTask step may not be exactly accurate for its mapped sentence regarding the same video”.
> 3. Our evaluation setup is better suited to the task:
>    - They only evaluate step grounding recall, ignoring non-groundable steps, while our annotations enable the computation of mAP (precision and recall).
>    - We propose a metric tailored to article grounding.
>    - We include both seen and unseen test sets.
>
> We have clarified those points in the related work section (see updated manuscript, lines 95-101)
>
> **Q4. It would be interesting to validate the potential of HT-Step on pre-training. For example, pre-training the model on HT-step and evaluating (e.g. Finetuning or zero-shot testing) on downstream dataset (e.g. CrossTask).**
>
> We thank the reviewer for this suggestion. To showcase the potential of HT-Step for pre-training models we evaluate it in a zero-shot fashion on CrossTask.
> We pre-trained the ActionFormer-T model on the ASR captions of HowTo100M, then fine-tuned the model with the HT-Step annotations and evaluate it on CrossTask, following established protocol. It is clear that the fine-tuning results in a huge boost in performance (+11.4% absolute improvement) but also surpasses previous SOTA on this benchmark by +6.5%.. Repeating the same experiment using COIN annotations instead, results in much lower performance, which further corroborates the quality of our annotations.
>
> | Method                           | Avg. Recall |
> |----------------------------------|-------------|
> | Zhukov                           | 22.4        |
> | HT100M                           | 33.6        |
> | VideoCLIP                        | 33.9        |
> | MCN                              | 35.1        |
> | DWSA                             | 35.3        |
> | MIL-NCE                          | 40.5        |
> | VT-TWINS                         | 40.7        |
> | UniVL                            | 42.0        |
> | ActionFormer-T (HowTo100M)       | 37.1        |
> | ActionFormer-T (HowTo100M + COIN)| 41.4        |
> | ActionFormer-T (HowTo100M + HT-Step)| **48.5**    |
>
> We have included the above experiments and analysis in the main paper (see lines 291 - 305 and tables 5,6 in updated manuscript).

---

### Official Review · Reviewer_XtLe · 2023-07-20
**Review for HT-Step**

**Rating:** 7
**Confidence:** 4
**Correctness:** The claims are all seemingly correct.
**Clarity:** It's clearly written.

**Strengths:**

1. Large-scale dataset: The paper introduces a large-scale dataset of instructional videos and instructional articles, which significantly surpasses existing labeled step datasets in terms of scale, number of tasks, and richness of natural language step descriptions. This dataset provides a valuable resource for training and evaluating models for aligning instructional articles with how-to videos.

2. Benchmark for temporal article grounding: The paper proposes a benchmark for the task of temporal article grounding, where the goal is to temporally localize the steps demonstrated in a video and reject the ones not shown. By providing this benchmark, the authors enable the development and evaluation of language-based temporal alignment models that can effectively ground textual descriptions of steps in instructional articles, even for unseen tasks.

3. Taxonomy of steps: The authors leverage wikiHow articles to guide the annotation process and create a taxonomy of steps fitting the data. This taxonomy provides a structured framework for organizing and categorizing the steps described in instructional articles, enhancing the understanding and alignment of instructional content with corresponding video segments.

**Additional Feedback:**

N/A

**Documentation:**

It has sufficient documentation of the data and annotations.

**Limitations:**

I believe the contribution of this work is sufficient. I do not have comments for the limitations.

**Opportunities For Improvement:**

N/A

**Relation To Prior Work:**

It's clearly discussed.

**Summary And Contributions:**

Procedural videos and instructional articles are popular resources for learning various skills. However, while videos provide visual demonstrations, they lack structured step annotations. On the other hand, instructional articles offer detailed explanations but lack dynamic demonstrations. This paper addresses the need for aligning instructional articles with how-to videos to enhance the learning experience.

Previous methods have focused on understanding procedural videos and temporal grounding tasks, but they lack comprehensive step annotations. The authors of this paper aim to bridge this gap by introducing HT-Step, a dataset with labeled step segments on instructional videos. They leverage wikiHow articles to guide the annotation process and create a taxonomy of steps fitting the data. The dataset includes over 116k temporal step labels, significantly surpassing existing datasets in terms of scale, taxonomy granularity, and scope.

The motivation behind this research is to provide a benchmark for the task of temporal article grounding, where the goal is to temporally localize the steps demonstrated in a video and reject the ones not shown. By introducing this benchmark, the authors hope to encourage the development of language-based temporal alignment models that can effectively ground textual descriptions of steps in instructional articles, even for unseen tasks.

---

> ### Author Response · Authors · 2023-08-22
> **Rebuttal**
>
> We thank the reviewer for the thorough review and comments.

---

### Official Review · Reviewer_6wqD · 2023-07-22
**Review of HT-Step: Aligning Instructional Articles with How-To Videos**

**Rating:** 5
**Confidence:** 3
**Clarity:** The paper is well written and easy to…

**Strengths:**

1. The proposed dataset is large-scale. The step annotations  are much larger than pervious datasets.

2. The paper proposes a challenging benchmark for the task of temporal article grounding, and evaluate SOTA methods on it.

3. The details of dataset collection, annotation and statistics are well demostrated.

**Additional Feedback:**

The authors should clearly answer my concerns about the soundness of the dataset construction above.

**Correctness:**

The soundness of the dataset construction should be concerned.  The annotations are the alignment of texts and video segments instead of directly annotation. Thus the annotation quality is not high enough.

**Documentation:**

The details of the data collection, organization and statistics are demonstrated. The informantion about the evaluation benchmarks is also provided.

**Ethics:**

There is no ethical concerns about the provided dataset.

**Limitations:**

The authors addressed the limitations in the paper. They claimed that  models trained on the dataset may exhibit biases towards the specific activities included in the dataset. Therer is no potential negative societal impact of their work.

**Opportunities For Improvement:**

1. It seems the only significance of the dataset is its large scale. The tasks that the dataset can apply already exist and there is no special use of the dataset compared with previous datasets.


2. The accuracy of the annotation of this dataset is worth considering. The annotations are not directly labeled by annotators. The annotations are existing wikiHow texts that aligned to the video segments. Thus, I suspect that the textual description is not entirely accurate. In the paper, the partial matchs means that some sub-steps in the step headline are skipped, some ingredients are not used, or that a step is interrupted and continued later. And the 73% of the annotations are partial matchs. Thus I think the quality of annotation is not high enough.

**Relation To Prior Work:**

There is a clear comparison between the proposed dataset and previous datasets in Table 1. However, there is no clear text description to discuss how this work differs from previous contributions. It is better to add a paragrapth about it.

**Summary And Contributions:**

The paper introduces HT-Step, a large-scale dataset of temporal annotations of instructional article steps in cooking videos. It leverages wikiHow articles and HowTo100M videos to create a rich and diverse collection of step labels and descriptions. The paper also proposes a benchmark for the task of temporal article grounding, which aims to align the steps of an article with the segments of a video. The paper evaluates several state-of-the-art methods and baselines on this task, and shows that combining weakly supervised pretraining with strongly supervised finetuning leads to the best performance. The paper also analyzes the properties and challenges of the dataset, such as step ordering variation, partial matches, and non-groundable steps. The paper hopes to advance the field of procedural activity understanding by releasing the annotations and providing evaluation protocols.

---

> ### Author Response · Authors · 2023-08-22
> **Author rebuttal**
>
> We thank the reviewer for the thorough review and comments.
>
> **Q1. Seems that the only significance of the dataset is its large scale. The tasks that the dataset can apply already exist and there is no special use of the dataset compared with previous datasets.**
>
> There is no other existing dataset with temporal segments for training and evaluating on the article grounding task. This task was introduced in [WSAG, Chen et al, 2022], however this work was limited to weakly-supervised training due to the lack of a strongly labeled dataset, and used CrossTask as a proxy dataset for evaluation, with noisy mapping between article steps and CrossTask labels. In more detail,
> - For training they use only the video-level activity labels instead of temporal segments, resulting in models that perform rather poorly, e.g., worse than the MIL-NCE model trained on ASR.
> - For evaluation they resort to using the CrossTask dataset by employing manually mapping CrossTask labels to wikiHow steps. As they describe: “When manually mapping the “step” in CrossTask to the “sentence” in the wikiHow article, we found it not always be one-to-one perfect mapping. In a few cases, multiple sentences may refer to a single step or multiple steps may refer to a single sentence. Thus, the original ground-truth annotations for each CrossTask step may not be exactly accurate for its mapped sentence regarding the same video”. Due to this limitation, they could only evaluate step grounding recall.
>
> The above further confirms that there is currently both a gap and a demand for temporal segment annotations of instructional article steps in the existing dataset landscape, which we argue HT-Step can timely cover.
>
> **Q2. I suspect that the textual description is not entirely accurate, [...] Thus I think the quality of annotation is not high enough.**
>
> Our dataset contains high-quality annotations of all the temporal segments that are related to any of the sentences of an article step - A rigid QA process has been applied on the vast majority of the samples that made it into the test sets (detailed in lines 148-154 of the updated manuscript).
>
> The fact that the textual description (headline + paragraph) of an article step may not always perfectly align with the contents of the video is an **inherent challenge of the article grounding task**. A common taxonomy among multiple videos was one of the requirements and design choices. This is in contrast to vanilla natural language description grounding, where textual descriptions completely match the video, or action detection, where a temporal segment completely matches its label.
>
> In addition,
> - We have demonstrated through experimentation with multiple architectures that our annotations enable the learning of strong article grounding models.  Finetuning the weakly-supervised model on our labels leads to a dramatic absolute gain of +23.9% (29.8 vs 5.9) in mAP on seen steps and +15.5% on unseen steps (see Table 2).
> - We present here zero-shot action localization results on the established CrossTask benchmark. Starting from an ActionFormer-T pre-trained on the ASR captions of HowTo100M, we finetune it on HT-Step, and then evaluate on CrossTask. We report below the performance using the standard Average Recall metric. It can be seen that fine-tuning on HT-Step yields a huge boost in performance compared to the pretraining on ASR (+11.4% absolute improvement) and leads to a new SOTA on this benchmark (+6.5%).
> - Repeating the same experiment using COIN annotations instead, results in much lower performance, which further corroborates the quality of our annotations. We conjecture this is due to the richness of the HT-step descriptions compared to the laconic COIN labels.
>
> | Method                           | Avg. Recall (%) |
> |----------------------------------|-------------|
> | UniVL  - SOTA                          | 42.0        |
> | ActionFormer-T (HowTo100M)       | 37.1        |
> | ActionFormer-T (HowTo100M + COIN)| 41.4        |
> | ActionFormer-T (HowTo100M + HT-Step)| **48.5**     |
>
> To conclude, although HT-Step is tailored towards article grounding - representing the first large-scale dataset for this task - it can also be a rich resource for training general natural language localization or action detection models.
>
> **Q3. There is no clear text description to discuss how this work differs from previous contributions. It is better to add a paragraph about it.**
>
> We discuss the main differences from prior work in the related work section (lines 94-111, see updated manuscript). We briefly summarize them here:
>  - We are focused on grounding entire articles instead of localizing single text queries.
>  - HT-step includes ungroundable steps (vs paragraph grounding, where each sentence is groundable).
>  - We include both seen and unseen evaluation settings to encourage research in zero-shot article grounding.
>  - Annotation scale and diversity are significantly higher than prior work (summary in Table 2).

---

### Author Response · Authors · 2023-08-22
**Author Rebuttal**

We thank the reviewers for their thorough reviews and insightful feedback. Following the reviewers' suggestions we have updated the main paper, adding additional relevant content, which is marked in blue text. Concretely this includes:
- Clarifications of the contributions compared to related work (lines 95-101)
- A schematic of the annotation pipeline  (Figure 2)
- Extra experimental results showcasing the value of HT-Step as a pre-training dataset for step localization tasks (Tables 5,6, lines 291-305).

We provide individual answers to the reviewers' comments and welcome further discussion as well as requests for additional clarifications.

---

### Decision · Program_Chairs · 2023-09-22

**Decision:**

Accept (Poster)

**Comment:**

The paper receives mostly positive reviews except one. The positive reviews acknowledges the size and clever use of wikiHow data. The only one negative reviewer raised questions about novelty of the task, not only the authors provide comprehensive answers to them and the reviewer did not actively respond to the authors' response but also the concern is not quite valid -- the task that the dataset is addressing is not necessarily novel. Given the value of the proposed dataset, the AC recommends to accept the submission to NeurIPS 23 DB track.